# SEMSTAMP: A SEMANTIC WATERMARK WITH PARAPHRASTIC ROBUSTNESS FOR TEXT GENERATION

## ABSTRACT

Existing watermarking algorithms are vulnerable to paraphrase attacks because of their token-level design. To address this issue, we propose SEMSTAMP, a robust sentence-level semantic watermarking algorithm based on locality-sensitive hashing (LSH), which partitions the semantic space of sentences. The algorithm encodes and LSH-hashes a candidate sentence generated by an LLM, and conducts sentence-level rejection sampling until the sampled sentence falls in watermarked partitions in the semantic embedding space. A margin-based constraint is used to enhance its robustness. To show the advantages of our algorithm, we propose a "bigram" paraphrase attack using the paraphrase that has the fewest bigram overlaps with the original sentence. This attack is shown to be effective against the existing token-level watermarking method. Experimental results show that our novel semantic watermark algorithm is not only more robust than the previous state-of-the-art method on both common and bigram paraphrase attacks, but also is better at preserving the quality of generation.

## 1 INTRODUCTION

Large language models (LLMs) such as GPT-4 (OpenAI, 2023) generate realistic text and follow instructions given a user-specified prompt. However, such capabilities also increase risks of misusing LLMs such as generating misinformation, impersonating, copyright infringements, and more (Weidinger et al., 2021; Pagnoni et al., 2022; Crothers et al., 2023; Ippolito et al., 2022). Therefore, methods for detecting machine-generated text (Jawahar et al., 2020; Ippolito et al., 2020; Mitchell et al., 2023, *i.a.*) as well as regulating its proliferation (House, 2023) is a crucial step towards reducing harms. This work focuses on algorithms for *watermarked generation*—an approach which facilitates the detection of machine-generated text by adding algorithmically detectable signatures during LLM generation which are imperceptible to humans (Atallah et al., 2001).

In a recent and impactful work, Kirchenbauer et al. (2023a) propose a watermark algorithm that pseudo-randomly partitions the vocabulary into a "green list" and a "red list" based on the hash of the last generated token, and injects the watermark by biasing the LLM to generate more green list tokens. Although this watermarking algorithm is efficient, follow-up work has shown that corrupting the generated text, especially paraphrasing, could weaken its robustness (Krishna et al., 2023; Sadasivan et al., 2023; Kirchenbauer et al., 2023b).

In this work, we propose SEMSTAMP, a *semantic watermark algorithm* that is robust to sentence-level paraphrase attacks (§2.2). Depicted in Figure 1, our core intuition is that while paraphrasing alters the surface-form tokens, the sentence-level semantics are not changed. Thus, instead of partitioning the vocabulary, our watermark operates on the semantic space of sentence embeddings, partitioned by locality-sensitive hashing (LSH; Indyk & Motwani, 1998; Charikar, 2002). As a key component, we use a paraphrase-robust sentence encoder trained with contrastive learning (CL; Wieting et al., 2022).

To test the robustness of watermarking algorithms, we further develop a novel attack method that minimizes bigram overlap during paraphrasing, namely the bigram paraphrase attack (§2.3). Experimental results (§3) demonstrate that our proposed semantic watermarking remains effective while token-level watermarks suffer significantly from the bigram attack.

We summarize our main contributions as follows. First, we propose a sentence-level semantic watermark for LLMs and show that it is robust to paraphrasers and more quality-preserving than

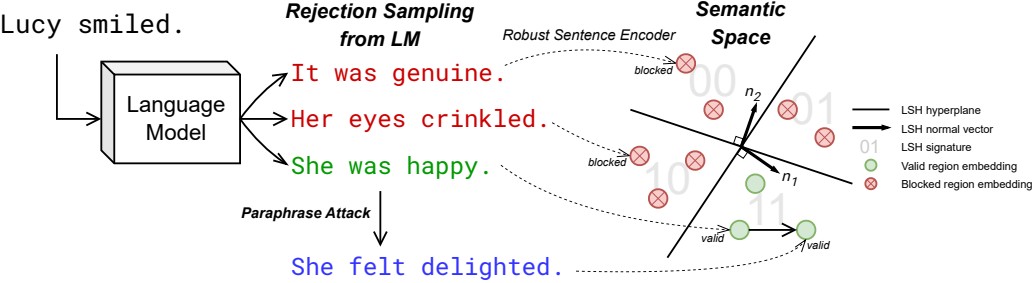

Figure 1: An overview of the proposed SEMSTAMP algorithm. The watermark is injected by mapping candidate sentences into embeddings through a robust sentence encoder, dividing the semantic space through locality-sensitive hashing, and rejection sampling from the LM to generate sentences with valid region embeddings.

a token-level watermark algorithm. Second, we develop a novel attack method for watermark algorithms, the bigram paraphrase attack, which can effectively weaken token-level watermarking but only poses a minor degradation to our semantic watermark. Third, we fine-tune a paraphrase-robust sentence encoder with a contrastive learning objective and develop a rejection margin constraint to enhance the paraphrastic robustness of our semantic watermark algorithm.[1]

## 2 APPROACH

### 2.1 PRELIMINARIES

**Text Generation from Autoregressive LMs** An autoregressive LM, denoted by $P_{LM}$, models the conditional distribution of the next token over the vocabulary $V$. Given a token history $w_{1:t} = w_1, \ldots, w_t$ where each token $w_i \in V$, the next token is generated by sampling $w_{t+1} \sim P_{LM}(\cdot|w_{1:t})$. A text sequence is generated by repeatedly sampling from the conditional distribution in a left-to-right fashion. We also introduce a sentence-level notation: $s^{(t+1)} \sim P_{LM}(\cdot|s^{(1)} \ldots s^{(t)})$ refers to the sampling of the next sentence given sentence history $s^{(1)} \ldots s^{(t)}$.

**Token-Level Watermarking and its Susceptibility to Paraphrase Attacks** Kirchenbauer et al. (2023a) proposes a watermark that is injected at the token level. At each time step of the generation, the vocabulary $V$ is pseudorandomly partitioned into a "green list" and a "red list". The random seed for partition is computed by a hash of the previously generated token. A globally fixed bias parameter $\delta > 0$ is added to the logit of each green list token so that the LM is induced to generate more green list tokens. The watermark is detected by conducting one proportion $z$-test (detailed in §B) on the number of green list tokens in the generated text.

Because of the token-level nature of the watermark algorithm, perturbing a token $w_t$ in a generated sequence $w_{1:T}$ through paraphrasing would change the green list for token $w_{t+1}$. As a result, a green token $w_{t+1}$ could be considered red after the green list has changed, which undermines the detectability of the watermark (Krishna et al., 2023). Moreover, because the watermark changes logits directly, it can degrade the quality of generated text (Fu et al., 2023).

**Locality-Sensitive Hashing** We will use LSH (Indyk & Motwani, 1998) to partition the semantic embedding space. It hashes similar inputs into similar signatures, thereby reducing the dimensionality and providing a similarity measure for a high-dimensional input space $\mathbb{R}^h$. Given an LSH dimension $d$, we adopt the cosine-preserving method from Charikar (2002) which produces a $d$-bit binary signature through random hyperplane projections, and each hyperplane is represented by a random normal vector $n^{(i)}$ drawn from the $h$-dimensional Gaussian distribution.[2] The LSH signature for an embedding vector $v \in \mathbb{R}^h$ is then determined by the sign of the dot product between the candidate vector and the normal vectors:

$$\text{LSH}_i(v) = \mathbb{1}(n^{(i)} \cdot v > 0), \tag{1}$$

where $\mathbb{1}(\cdot)$ is the indicator function, $\text{LSH}_i : \mathbb{R}^h \mapsto \{0, 1\}$ gives the $i$-th digit signature, and $\text{LSH}(v) = [\text{LSH}_1(v)||\ldots||\text{LSH}_d(v)]$ is the concatenation of all $d$ digits.

---

[1]Our code, model, and data will be released in the public version of this manuscript.

[2]Normal vector $n^{(i)} \in \mathbb{R}^h$ represents the hyperplane that is orthogonal to $n^{(i)}$ and passes through the origin.

---

**Algorithm 1** SEMSTAMP

---

**Input:** language model $P_{\text{LM}}$, prompt $s^{(0)}$, number of sentences to generate $T$.
**Params:** sentence embedding model $M_{\text{embd}}$ with embedding dimension $h$, maxout number $N_{\text{max}}$, margin $m > 0$, valid region ratio $\gamma \in (0, 1)$, LSH dimension $d$, a large prime number $p$.
**Output:** generated sequence $s^{(1)} \ldots s^{(T)}$.

**procedure** SEMSTAMP
    **init** LSH$(\cdot)$, randomly initialize $d$ LSH hyperplanes, represented by the normal vectors $n^{(1)} \ldots n^{(d)} \in \mathbb{R}^h$, to create $2^d$ semantic subspaces.
    **for** $t = 1, 2, \ldots, T$ **do**
        1.  Compute the LSH signature of the previously generated sentence, SIG$(s^{(t-1)})$, and use $\left[\text{SIG}(s^{(t-1)})\right]_{10} \cdot p$ as the seed to randomly divide the space of signatures $\{0, 1\}^d$ into a "valid region set" $G^{(t)}$ of size $\gamma \cdot 2^d$ and a "blocked region set" $R^{(t)}$ of size $(1 - \gamma) \cdot 2^d$.

        2.  **repeat** Sample a new sentence from LM, $s^{(t)} \sim P_{\text{LM}}(\cdot | s^{(1)} \ldots s^{(t-1)})$
            **until** the signature of the new sentence is in the "valid region set", SIG$(s^{(t)}) \in G^{(t)}$ and the margin requirement MARGIN$(s^{(t)}, m)$ is satisfied.
            **or** has repeated $N_{\text{max}}$ times
        3.  Attach the selected sentence $s^{(t)}$ to context.
    **end for**
    **return** $s^{(1)} \ldots s^{(T)}$
**end procedure**

**function** SIG$(s)$                                     ▷ Sentence $s$
    $v \leftarrow M_{\text{embd}}(s)$         ▷ Apply the embedding model to get sentence embedding $v$
    $c \leftarrow \text{LSH}(v)$         ▷ Discretize the embedding via LSH to produce signature $c$
    **return** $c$
**end function**

**function** MARGIN$(s, m)$                          ▷ Sentence $s$, margin $m$
    $v \leftarrow M_{\text{embd}}(s)$         ▷ Apply the embedding model to get sentence embedding $v$
    $x \leftarrow \min_{i=1,\ldots,d}\{|\cos(v, n^{(i)})|\}$     ▷ Compute the minimum absolute cosine similarity between $v$ and all LSH normal vectors $n^{(i)}$.
    **return** True **If** $x \geq m$ **Else** False
**end function**

---

## 2.2 SEMSTAMP: A SEMANTIC WATERMARK WITH PARAPHRASTIC ROBUSTNESS

We begin with a high-level overview of the SEMSTAMP algorithm. Our approach is motivated by the intuition that paraphrasing alters the surface-form tokens but preserves sentence-level semantics. We apply the watermark at the sentence-level semantic space (instead of the token-level vocabulary) to preserve the watermark under token changes. A core component of SEMSTAMP is a robust sentence encoder, denoted $M_{\text{embd}}$ in Algorithm 1. We fine-tune an off-the-shelf encoder with a contrastive learning objective (Wieting et al., 2022) for paraphrastic robustness.

At the initialization stage of SEMSTAMP, we partition the space of sentence embeddings $\mathbb{R}^h$ of $M_{\text{embd}}$, a representation of the sentence-level semantic space, with the LSH method introduced in §2.1. Concretely, we initialize the LSH : $\mathbb{R}^h \mapsto \{0, 1\}^d$ function by sampling normal vectors $n^{(1)} \ldots n^{(d)}$ to represent $d$ LSH hyperplanes, and treat the space of LSH signatures $\{0, 1\}^d$ as a natural partitioning of $\mathbb{R}^h$ into $2^d$ regions.

At each generation step, given a sentence history $s^{(0)} \ldots s^{(t-1)}$, we first produce the LSH signature of the previously generated sentence SIG$(s^{(t-1)})$. Next, we pseudorandomly divide the LSH partitions into a set of "valid" regions $G^{(t)}$ and a set of "blocked" regions $R^{(t)}$, where the masking is seeded by

$\textsc{Sig}(s^{(t-1)})$.[3] To produce the watermarked next sentence, we sample with rejection a new sentence $s^{(t)}$ from the LM until its embedding lies in the "valid" region within the semantic space.

Because a proper paraphrase should retain the meaning of the original sentence, we hypothesize that the LSH signature for a candidate sentence should rarely change after paraphrasing (Figure 4 provides empirical results). Therefore, the valid region partition for the next sentence would not change, ensuring the watermark is still detectable after the paraphrase attack. Below we explain each core component of SEMSTAMP in detail.

**Paraphrase-Robust Sentence Encoder**  A prerequisite for SEMSTAMP is a semantic embedding model to encode candidate sentences into sentence embeddings. We base our encoder on Sentence-BERT (SBERT; Reimers & Gurevych, 2019), a fine-tuned siamese network that produces sentence embeddings whose cosine similarity approximates the semantic textual similarity on the STS benchmark (Cer et al., 2017).

To enhance the encoder's robustness to paraphrase, we further fine-tune the SBERT model using contrastive learning following Wieting et al. (2022). For each sentence $s_i$ in a corpus, we first produce its paraphrase $t_i$ using an off-the-shelf paraphrasing model, Pegasus (Zhang et al., 2020).[4] Next, we sample a random sentence $t_i'$ from the corpus that is not a paraphrase of $s_i$ to serve as the negative example. The objective promotes the original sentence to be more similar to the paraphrase than the negative example by a margin of $\delta > 0$:

$$\min_{\theta} \sum_i \max\left\{\delta - f_\theta(s_i, t_i) + f_\theta(s_i, t_i'), 0\right\}, \tag{2}$$

where $f_\theta$ is the cosine similarity between the embedded sentences, $f_\theta(s, t) = \cos\big(M_\theta(s), M_\theta(t)\big)$, and $M_\theta$ is the encoder model with parameter $\theta$.

**Semantic Space Partitioning through LSH**  In the initialization stage of watermarked text generation, normal vectors $n^{(1)} \ldots n^{(d)}$ are randomly drawn from the $h$-dimensional Gaussian distribution in the semantic space of $\mathbb{R}^h$ to represent $d$ LSH hyperplanes. The hyperplanes are fixed during generation and detection to serve as the basis for partitioning. As introduced in §2.1, this induces a $d$-bit binary signature $\textsc{Lsh}(v)$ for a vector $v \in \mathbb{R}^h$. Consequently, we use each of the $2^d$ signatures $c \in \{0, 1\}^d$ to represent a region in the semantic space consisting of points with signature $c$.

During the generation of a new sentence $s^{(t)}$, we apply a watermarking "mask" on the semantic space by pseudorandomly partitioning the space of signatures $\{0, 1\}^d$ into a valid region set $G^{(t)}$ of size $\gamma \cdot 2^d$ and a blocked region set $R^{(t)}$ of size $(1 - \gamma) \cdot 2^d$, where $\gamma \in (0, 1)$ determines the ratio of valid regions. The masking is seeded by the LSH signature of the last sentence $s^{(t-1)}$ and thus varies for each sentence-step. Specifically, we convert the binary signature $\textsc{Sig}(s^{(t-1)})$ to decimal and use $[\textsc{Sig}(s^{(t-1)})]_{10} \cdot p$ (where $p$ is a large prime number) to seed the randomization. The condition for rejection sampling is that the LSH signature of the new sentence must fall into one of the valid regions, i.e., $\textsc{Lsh}(M_{\text{embd}}(s^{(t)}) \in G^{(t)}$.

**Margin-Based Constraint for Enhanced Robustness**  For robustness, the SEMSTAMP algorithm would need the LSH signature of the paraphrased sentence to be unchanged from the signature of the original sentence, i.e., for each LSH digit $i$, the sign of the dot product between the embedded sentence and the normal vector $n^{(i)}$ should not change before and after paraphrasing:

$$\mathbb{1}\big(n^{(i)} \cdot v_{\text{orig}} > 0\big) = \mathbb{1}\big(n^{(i)} \cdot v_{\text{para}} > 0\big), \forall i \in \{1, \ldots, d\}, \tag{3}$$

where $v_{\text{orig}} = M_{\text{embd}}(s^{(t)})$ and $v_{\text{para}} = M_{\text{embd}}(G(s^{(t)}))$ are the embeddings for the original and paraphrased sentences, respectively, and $G$ is the paraphraser.

Empirically, we found the robustness from contrastive learning (Eq. 2) is not strong enough to preserve consistent LSH signature under paraphrasing. Therefore, we add an additional rejection sampling requirement that the sampled sentence $s^{(t)}$ must have the absolute value of cosine similarity with each normal vector $n^{(i)}$ larger than a margin $m > 0$:

---

[3]Kirchenbauer et al. (2023a) use "green/red" for vocabulary split. Instead, we adopt "valid/blocked" as the terminology for semantic region partition to be more accessible.

[4]https://huggingface.co/tuner007/pegasus_paraphrase

$$\min_{i=1,\ldots,d} \big| \cos(n^{(i)}, v_{\text{orig}}) \big| > m. \tag{4}$$

Visually, this is akin to rejecting sentences whose embeddings lie near the boundaries of an LSH hyperplane. We provide an illustration in Figure 2. In our experiments (§3), we show that this margin-based rejection requirement can effectively increase the LSH signature robustness under paraphrasing.

### 2.3 THE BIGRAM PARAPHRASE ATTACK

Because existing token-level watermark algorithms hash the last generated token to determine the green/red list split for the vocabulary (Kirchenbauer et al., 2023a), the change of token at position $t$ would affect the watermark of position $t + 1$. Due to this design choice, we hypothesize that token-level watermarks might be especially sensitive to bigram (two adjacent tokens) perturbation.

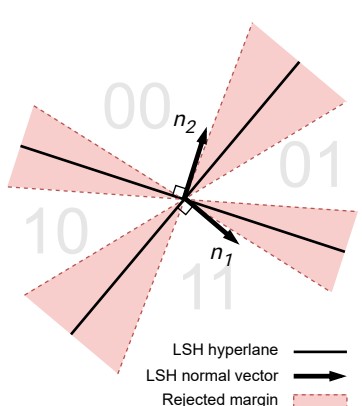

Figure 2: An illustration for margin-based rejection. Sentence embeddings at LSH hyperplane boundaries are rejected (highlighted in red).

Motivated by this issue, we propose and explore the bigram paraphrase attack, a simple yet effective variant of the basic sentence-level paraphrase attack. Specifically, given a neural paraphrase model $G$, the standard method for paraphrasing is using beam search to decode a top-ranking sequence $s'$ given the original sentence $s$. To conduct the bigram attack, instead, we first decode a large number of top-raking sequences $s'_1 \ldots s'_k$ from beam search, obtaining $k$ paraphrase candidates. Next, we select the candidate that has the smallest bigram overlap with the original sentence. Moreover, to preserve the paraphrasing quality, we constrain the paraphrase attack with BERTScore (Zhang et al., 2019) between paraphrases and original sentences:

$$
\begin{aligned}
s' = \;& \underset{x \in \{s'_1, \ldots, s'_k\}}{\arg\min} \quad \mathcal{B}(x, s), \\
& \text{subject to} \quad \mathcal{S}(s'_1, s) - \mathcal{S}(x, s) \le \Delta \cdot \mathcal{S}(s'_1, s),
\end{aligned}
\tag{5}
$$

where $s$ denotes the original sentence, $\mathcal{B}(x, s)$ is a simple counting of overlapped bigrams between sequences $x$ and $s$, $\mathcal{S}(x, s)$ denotes the BERTScore between sequence $x$ and $s$, and $\Delta$ is the BERTScore threshold ratio. See Figure 5 for an example in action.

## 3 EXPERIMENTS

### 3.1 EXPERIMENTAL SETUP

**Datasets and Metrics** We use OPT-1.3B as our autoregressive LM to generate texts and semantically watermark generations with a finetuned Sentence-BERT model. The RealNews subset of the c4 dataset (Raffel et al., 2020) is used for both Sentence-BERT finetuning and evaluation of watermark algorithms. We analyze the detection results and generation quality on 500 samples per parameter combination. 200 samples are used for development.

To evaluate the effectiveness of watermarked detection, we utilize binary classification metrics **AUROC** (area under the receiver operating characteristic curve) and the true positive rate when the false positive rate is 5% (**TP@FP=5%**), i.e., the percentage of machine-generated text (the "positive" class in the classification setting) that is correctly detected when 5% of human texts (the "negative" class) are misclassified as machine-generated texts. A piece of text is classified as machine-generated when its $z$-score exceeds a threshold chosen based on a given false positive rate, which we explain in detail in §B. Note that differing from the baseline algorithm in Kirchenbauer et al. (2023a), our algorithm treat sentences as the unit during $z$-score computation.

To evaluate generation quality, we measure the perplexity (**PPL**) with OPT-2.7B (Zhang et al., 2022). Diversity is measured with trigram text entropy (Zhang et al., 2018) (**Ent-3**), i.e., the entropy of the trigram frequency distribution of the generated text. We also use the seq-rep-3 (**Rep-3**) metric from Welleck et al. (2020), which measures the proportion of repeated trigrams in generated text. We measure the quality of paraphrases using **BERTScore** (Zhang et al., 2019) between original generations and their paraphrases.

| Paraphraser | Algorithm | AUROC ↑ | TP@FP=5% ↑ | BERTScore (para.) |
|---|---|---|---|---|
| No Paraphrase | Baseline | 0.998 | 0.992 | – |
| | SSTAMP | 0.998 | 0.998 | – |
| Pegasus | Baseline | 0.971 | 0.935 | 0.710 |
| | SSTAMP | **0.981** | **0.985** | 0.691 |
| Pegasus-bigram | Baseline | 0.944 | 0.842 | 0.678 |
| | SSTAMP | **0.974** | **0.980** | 0.663 |
| Parrot | Baseline | 0.915 | 0.787 | 0.565 |
| | SSTAMP | **0.926** | **0.817** | 0.537 |
| Parrot-bigram | Baseline | 0.875 | 0.658 | 0.554 |
| | SSTAMP | **0.928** | **0.834** | 0.551 |
| GPT3 | Baseline | 0.932 | 0.787 | 0.609 |
| | SSTAMP | **0.934** | **0.822** | 0.629 |
| GPT3-bigram | Baseline | 0.898 | 0.647 | 0.573 |
| | SSTAMP | **0.915** | **0.929** | 0.592 |

Table 1: Detection results under different paraphraser settings. Baseline refers to the watermarking algorithm in Kirchenbauer et al. (2023a), and SSTAMP refers to the proposed SEMSTAMP algorithm. The proposed SEMSTAMP algorithm, referred in the table as SSTAMP, is more robust than the baseline on a number of paraphrasers and both the regular and bigram paraphrase attacks.

**Training and Generation**    For contrastive learning of Sentence-BERT, we paraphrase 8k paragraphs of the C4-RealNews dataset (Raffel et al., 2020) using the Pegasus paraphraser (Zhang et al., 2020) through beam search with 25 beams. We then fine-tune a Sentence-BERT model [5] with an embedding dimension $h = 768$ on this subset for 3 epochs with a learning rate of $4 \times 10^{-5}$, using contrastive learning objective (Eq. 2). We set the contrastive learning margin $\delta = 0.8$ which is tuned from the dev set.

During generation, we use OPT-1.3B (Zhang et al., 2022) as our base model and conduct sampling at a temperature of 0.7 following Kirchenbauer et al. (2023a) with a repetition penalty of 1.03. We set 32 as prompt length and generate to various lengths, with 200 tokens being our default length.

In the paraphrase attack phase, we paraphrase SEMSTAMP generations and baseline watermark generations and compare their post-hoc detection rates. We sample at a LSH dimension $d = 2$ and a valid region ratio $\gamma = 0.25$. We also set our rejection margin $m = 0.02$. See §3.3 for the impact on hyperparameter choices.

**Paraphrase Attack**    For paraphrase attack experiments, we paraphrase watermarked generations sentence by sentence using the Pegasus paraphraser (Zhang et al., 2020), the Parrot paraphraser in Sadasivan et al. (2023), and GPT-3.5-Turbo (OpenAI, 2022). We use beam search with 25 beams for both Pegasus and Parrot. For GPT-3.5-Turbo, we provide the sentences before the current sentence as the context and prompt the model to paraphrase via the OpenAI API. [6] We provide a detailed description of prompts in §D.

To implement the bigram paraphrase attack, we prompt the GPT-3.5-Turbo to return 10 paraphrases of the same sentence. For the Pegasus and Parrot paraphrasers, we select the candidate sentence with the least bigram overlap among the 25 beams from beam-search, subject to a BERTScore constraint of dropping no more than 10% of the score from the first beam. For GPT-3.5-Turbo, the paraphrase sample with the highest BERTScore is treated as the first beam.

## 3.2 RESULTS

We first generate texts with different watermark algorithms, and then paraphrase the generations to attack on their watermarks. Table 1 shows detection results under different paraphrasers and the bigram attack at generation length 200. It is shown that SEMSTAMP is more robust to paraphrase

---

[5]sentence-transformers/all-mpnet-base-v1
[6]https://platform.openai.com/playground/

Figure 3: Detection results (AUROC) under different generation lengths. SEMSTAMP is more robust than the baseline (Kirchenbauer et al., 2023a) across length 100-400 tokens.

attacks than the baseline watermark across the Pegasus, Parrot, and GPT-3.5-Turbo paraphrasers, as measured by AUROC and TP@FP=5%. Although we only fine-tune the Sentence-BERT model on data from the Pegasus paraphraser, SEMSTAMP algorithm generalizes its robustness to different paraphrasers.

The bigram paraphrase attack is quite effective against the token-level baseline algorithm, whereas SEMSTAMP is relatively unaffected. For instance, Pegasus bigram attack lowers the baseline AUROC by 5.3%, whereas SEMSTAMP only decreases by 2.4%. Furthermore, the BERTScore for bigram paraphrase does not change drastically compared to the regular paraphrases, showing that the bigram paraphrase attack still preserves paraphrase quality due to the BERTScore constraints we add.

Table 2 compares quality metrics of non-watermarked generations with the baseline watermark and SEM-STAMP generations. While SEMSTAMP generation perplexity is on par with the vanilla model, the baseline watermark notably degrades the quality due to the probability shift in selecting valid tokens. On the other hand, since SEMSTAMP is sentence-level, it does not disrupt token selections and preserves the generation quality. Figure 5 further shows examples of SEMSTAMP generations and the bigram paraphrase attack. Compared to the non-watermarked (vanilla) text, the sentences are equally coherent and contextually sensible.

|  | *PPL*↓ | *Ent-3*↑ | *Rep-3*↓ |
|---|---|---|---|
| No watermark | 6.995 | 12.43 | .14 |
| Baseline | 8.455 | 12.33 | .19 |
| SSTAMP | **6.862** | 12.04 | .20 |

Table 2: Quality evaluation results. SEM-STAMP preserves the quality of generated text.

The two watermark algorithms also maintain the same level of text diversity and n-gram uniqueness compared to the vanilla generation, as measured by trigram entropy and n-gram frequency tests.

Figure 3 highlights that SEMSTAMP is robust to both regular and bigram paraphrase attacks across different generation lengths as measured by the number of tokens. SEMSTAMPhas consistently higher AUROC than the baseline (Kirchenbauer et al., 2023a), especially leading by greater margins facing the bigram paraphrase attacks.

Applying the masking of semantic space partitions and the rejection margin, SEMSTAMP makes a trade-off between watermark detection accuracy and generation speed. For our current hyperparameter setting, 14.6 sentences are needed on average to sample one valid sentence. As we explain in §4, this limitation can be mitigated if we conduct next-sentence sampling in parallel.

## 3.3 ANALYSIS

Figure 4 shows that increasing margin size $m$ will increase the consistency of LSH signatures (*LSH consistency*), i.e., the ratio of sentences that remain in the same valid region after being paraphrased. A higher rejection margin will ensure the sampled generations are further away from the region boundary, thus less likely to shift to a different region after paraphrasing. However, a larger margin will result in a slower generation speed, and we find $m = 0.02$ works well empirically.

We also compare the LSH consistency between off-the-shelf Sentence-BERT and its fine-tuned version by contrastive learning in Figure 4. It is shown that fine-tuning the encoder on Pegasus paraphrased data improves the LSH consistency across different margins.

## 4 LIMITATIONS AND DISCUSSION

---

**Prompt:** Marketers are employed in the public sector as well as the private sector.
**Vanilla Generation:** In the public sector, their job is to make sure that the government fulfills its goals. In the private sector, their job is to make sure that their company makes money. One of the primary tasks of marketers in the public sector is to keep customers updated on government services and programs.
**SEMSTAMP:** If you're a marketer, I'm sure you've heard of the term "branding." It's an important practice to understand how a brand is created and maintained. It's not the same thing as a marketing campaign, but it is a good starting point.

**Pegasus Paraphrase:** I'm sure you've heard of the term "branding." Understanding how a brand is created and maintained is an important practice. It isn't the same as a marketing campaign, but it is a good starting point.
**Pegasus Bigram Paraphrase:** I'm pretty sure you've heard of branding. Understanding how a brand is created is important. It's not a marketing campaign, but it's a good start.

---

Figure 5: Generation Examples. Vanilla refers to the original model without adding any watermark. Paraphrase examples are based on SEMSTAMP generations. Additional examples are presented in Figure 6 in the Appendix.

**Robustness to Stronger Attacks** Since SEMSTAMP operates on the sentence level, it is not robust against attacks on the inter-sentence level. For example, a recently proposed paraphraser Dipper (Krishna et al., 2023) includes sentence reordering. Our algorithm is also less effective when the machine text is embedded in a relatively large portion of human text. We leave the exploration of stronger attacks to future work.

**Semantic Constraint from LSH** While the LSH partitioning divides the full semantic space into sub-regions, enforcing the "valid region" requirement during generation may potentially reduce the generation flexibility. Interestingly, we use a small LSH dimension ($d = 2$) and we do not observe a visible quality degradation. A potential explanation is that with a smaller LSH dimension, the valid partition also becomes larger, which does not impose a strong semantic constraint and provides enough diversity for generations, as we found in our experiments (§3.2).

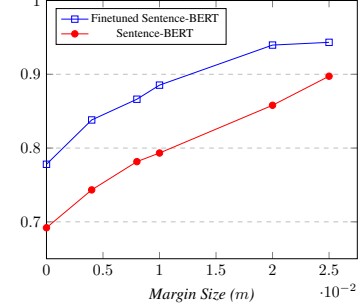

Figure 4: Effects of rejection margin and contrastive fine-tuning.

**Speed** Due to the nature of rejection sampling, text generation with SEMSTAMP is slower than non-watermarked generation by a factor of 14.6 with LSH dimension $d = 2$ and margin $m = 0.02$ (§3.2), and by a factor of 4.28 when $d = 2$ and $m = 0$ (Table 3). However, since candidate sentences for rejection sampling have the same LM context, it is possible to conduct batch sampling of candidate next sentences, which speeds up watermarked generation while increasing the memory overhead. We see the additional computation cost for SEMSTAMP as a cost for robustness: adding the watermark on the semantic space trades-off speed for better detection accuracy under paraphrase attacks. Further, a potential mitigation is through sampling candidate sentences with multiple devices at the same time.

**Reverse Engineering** Since our sentence encoder and LSH hyperplanes are not public, it is not straightforward for a curious attacker to reverse engineer the configurations and we leave it for future work to explore. The difficulty of reverse engineering can also be increased by using a larger LSH dimension, while the watermark could be less robust to paraphrase attack.

**Bigram Paraphrase Attack Control** We control the "intensity" degree of bigram paraphrase attack by constraining the paraphrase candidate selection with a BERTScore constraint. Removing the constraint will more forcefully lower AUROC at the expense of paraphrase quality.

## 5 RELATED WORK

Machine-generated text detection, aiming at distinguishing language model-generated texts from human-written ones, can be approached by both post-hoc and proactive methods. Our focus, watermarked generation, belongs to the second category.

**Post-Hoc Detection of Machine-Generated Text**    In post-hoc methods, applying binary classification models is the most straightforward approach (Zellers et al., 2019; Jawahar et al., 2020; Liu et al., 2022; Mireshghallah et al., 2023). These methods are applicable to black-box generators but need sufficient corpus to fine-tune in advance.

Another type of post-hoc detection is based on probability-related statistics, including token likelihood (Gehrmann et al., 2019), rank (Solaiman et al., 2019), entropy (Ippolito et al., 2020), and likelihood gap at perturbation (Mitchell et al., 2023; Su et al., 2023). These methods have better interpretation but are reliable only with white-box access to generators. Very recently, Sadasivan et al. (2023) question the theoretical reliability of detection while Chakraborty et al. (2023) support detection is achievable.

**Watermarked Generation**    Watermarked generation is an emerging trend of proactive machine-generated text detection, which adds signatures via controlled generation to enable stable detection. As a seminal work, Kirchenbauer et al. (2023a) proposes a watermarking algorithm by adding token-level bias (reviewed in §2). Yoo et al. (2023) further embed multi-bit information into watermark and enhance robustness against corruption by preventing altering keywords and high syntactic dependency components. However, they watermark via word replacement after initial generation, which is further improved into one-stage watermarked generation by Wang et al. (2023). These works focus on word-level attacks and do not consider paraphrasing.

Very recently, Christ et al. (2023) propose a watermarking scheme that is computationally undetectable without the secret key in theory. Fu et al. (2023) consider semantic word similarity during watermarked generation. Liu et al. (2023a) propose a private watermark using separate neural networks respectively for generation and detection. Kuditipudi et al. (2023) preserve the original distribution of LM during watermarking. These existing works mainly focus on editing, cropping, corruption, and copy-paste attacks.

More related to our focus on paraphrase attack, Krishna et al. (2023) propose a retrieval-based method that requires saving all previously-generated sequences, and Kirchenbauer et al. (2023b) empirically show that the robustness of the baseline algorithm is decent under relatively long generation length. Contemporary to our work, Zhao et al. (2023) improve robustness via a cryptographic-free watermark without hashing previous tokens, which is more robust to editing and paraphrasing attacks. To the best of our knowledge, our work is the first to propose a sentence-level semantic watermark algorithm that is directly targeted against paraphrase attacks.

**Locality-Sensitive Hashing in NLP**    The application of locality-sensitive hashing (Indyk & Motwani, 1998; Charikar, 2002) in NLP dates back to Ravichandran et al. (2005), where LSH is used for high-speed noun clustering. Van Durme & Lall (2010) show that the LSH method of Charikar (2002) can enable fast approximated online computation of cosine similarity. Guu et al. (2018) use LSH to efficiently compute lexically similar sentences in a prototype-then-edit sentence generation model. Closely related to our work, Weir et al. (2020) generate semantically diverse sentences by conditioning a sequence-to-sequence model on the LSH signature of sentence embeddings.

To save space, we defer discussion on watermarking on copyright as well as contrastive learning in NLP to §A.

## 6    CONCLUSION

We introduce SEMSTAMP, a novel sentence-level semantic watermark for LLMs. The watermark is injected by mapping candidate sentences into embeddings with a paraphrase-robust encoder, partitioning the semantic space through LSH, and rejection sampling to generation sentences with valid region embeddings. Empirical results show that SEMSTAMP is not only robust to paraphrase attacks but also more quality-preserving than a token-level baseline watermark algorithm. Our proposed bigram paraphrase attack effectively weakens the token-level watermark while only causing minor performance deterioration to SEMSTAMP. We hope SEMSTAMP can serve as an effective tool for regulating the proliferation of machine-generated texts.

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

SUPPLEMENTAL MATERIALS

# A ADDITIONAL RELATED WORKS

**Watermarked Natural Language Data for Copyright**   Watermarked generation can be further applied for data copyright protection. Gu et al. (2022) embed backdoor trigger words as black-box watermarks into LLMs. Liu et al. (2023b) propose a novel watermark via backdoor-based membership inference, where backdoor watermarked texts poison unauthorized training models. Yao et al. (2023) focus on protecting the copyright of prompts through inserting the secret key into the prompt optimization stage.These works mainly apply watermark techniques for data copyright protections , whereas our work focuses on exploring the robustness of watermark against paraphrasing.

**Contrastive Learning in NLP**   Contrastive learning (Hadsell et al., 2006) aims at improving the distinguishability of representation by pulling over positive pairs and pushing off negative pairs. In the NLP domain, contrastive learning can be applied to sentence embedding (Logeswaran & Lee, 2018), and further used in downstream tasks like natural language inference (Li et al., 2022), understanding (Fang et al., 2020), reasoning (Klein & Nabi, 2020), classification (Choi et al., 2022) etc. Logeswaran & Lee (2018) apply unsupervised contrastive learning between current sentence candidates and context sentences to effectively learn sentence representation. Gao et al. (2021) further apply supervised contrastive learning in sentence embedding by using annotated pairs from natural language inference. Kim et al. (2021) propose a self-guided contrastive learning between embeddings from a fixed model and a fine-tuned model.

# B WATERMARK DETECTION

Kirchenbauer et al. (2023a) proposes to use a one-proportion z-test to detect watermarks, assuming the following null hypothesis:

$H_0$ : *The text is not generated (or written) knowing a watermarking green list rule.*

The null hypothesis is rejected when the z-score computed based on the number of green tokens in a piece of text $T$ exceeds a given threshold $M$:

$$z_{baseline} = \frac{N_G - \gamma N_T}{\sqrt{\gamma(1-\gamma)N_T}}, \qquad (6)$$

where $N_G$ denotes the number of green tokens, $N_T$ refers to the total number of tokens contained in the given piece of text $T$, and $\gamma$ is a chosen ratio of green tokens. During detection time, the number of green tokens in each piece of text will be counted. According to Eq. 6, a higher ratio of detected green tokens means a higher z-score, determining with more confidence that the text is machine-generated.

We adapt this one proportion z-test to SEMSTAMP, modifying the null hypothesis and using sentence as our basic unit:

$H_0$ : *The text is not generated (or written) knowing a rule of*
*valid and blocked partitions in the semantic space*

$$z_{\text{SEMSTAMP}} = \frac{S_V - \gamma S_T}{\sqrt{\gamma(1-\gamma)S_T}}, \qquad (7)$$

where $S_V$ refers to the number of valid *sentences*, $\gamma$ is the ratio of valid sentences out of the total number of sentences $S_T$ in a piece of text $T$.

During detection time, we first break a piece of texts into individual sentences and detect the number of valid sentences $S_V$ to calculate $z_{\text{SEMSTAMP}}$. We detect a machine-generated text when $z_{\text{SEMSTAMP}} > M_r$, where $M_r$ is located according to a given false positive rate $r$: We define machine-generated as the positive class in classical classification setting and non-machine-generated as the negative class. We iterate through a range of possible $m \in [0, 4.0]$ until there is a $M_r = m$ such that $r$ percentage of human (negative-class) texts is misclassified as machine-generated. For example, we let $r = 0.05$ for the TP@FP=5% metric in Table 1.

---

**Prompt:** The Airline Vikings were playing for the state football championship in 1972 against Neville on a bad night for football in Monroe.

**Vanilla Generation:** They started the game by kicking a field goal from the OB́rien Field goal post and then went on to score another touchdown. After the game, the Vikingścoach, Bill Toutant, said the team played "like Vikings." One long-time fan of the Vikings told me this story before he died, so this is my story.

**Baseline Watermark:** The Viking offense looked like an NFL outfit and the game was over by halftime, led by a 42-7 victory for the Vikings. The next season, the Airline Vikings traveled to Lexington High School and finished the regular season with a 5-4 record. The Vikings had to win the state championship to make the playoffs, but they lost their first game in the state semifinals, finishing with a 5-4 record.

**SEMSTAMP:** Neville went ahead 14-0 in the second quarter and had a 28-0 halftime lead. Despite the odds, Airline cut the deficit to 28-14 late in the third quarter. But Neville's defense held on and won the game 28-14.

**Pegasus Paraphrase:** In the second quarter, they went ahead 14-0 and had a 28-0 lead. Airline cut the deficit to 28-14 late in the third quarter. The defense held on and won the game.

**Pegasus Bigram Paraphrase:** In the second quarter, Neville jumped out to a 14-0 lead. The deficit was cut to 28 in the third quarter. The game was won by the defense of Neville.'

---

Figure 6: Additional Generation Examples. Vanilla refers to the original model without adding any watermark. Baseline Watermark refers to Kirchenbauer et al. (2023a). Paraphrase examples are based on SEMSTAMP generations.

| LSH Dim ($d$) | Average Number of Sentences Sampled ↓ | LSH Consistency ↑ |
|---|---|---|
| 2 | 4.28 | **.778** |
| 4 | 4.19 | .628 |
| 8 | 3.81 | .435 |
| 16 | **3.78** | .246 |

Table 3: Effects of Increasing LSH Dimensions at margin $m = 0.0$. The sampling rate is the average number of sentences sampled to produce one valid (watermarked) sentence.

## C EFFECT OF LSH DIMENSION $d$

In Table 3, we discover that fewer LSH dimensions will make a sentence more likely to stay in the same region after being paraphrased. We define LSH Consistency as the ratio of paraphrased sentences that have the same LSH signature as the original sentence over the total number of paraphrased sentences. A higher consistency ratio indicates better robustness.

Geometrically, when the LSH dimension is lower, there are fewer partitioned semantic regions, each having a larger space. A paraphrase will have a similar representation with its source sentence in the semantic space, which will be more likely to remain in the same semantic region if each region is larger.

On the other hand, lowering the number of LSH dimensions will also slightly increase the average number of sentences sampled to produce one valid sentence (Average Number of Sentences Sampled). We ultimately decide on a minor sacrifice in speed for the gain of accuracy and choose $d = 2$, the smallest possible dimension under $\gamma = 0.25$. We chose $\gamma = 0.25$ following Kirchenbauer et al. (2023a), where the authors show that larger green-list ratios will lower the $z$-score.

## D ADDITIONAL DETAILS

**Cosine Similarity** In §2.2, we slightly abuse the notation and use $\cos(\boldsymbol{x}, \boldsymbol{y})$ to denote the *cosine similarity* between two vectors $\boldsymbol{x}$ and $\boldsymbol{y}$. That is,

$$\cos(\boldsymbol{x}, \boldsymbol{y}) = \frac{\boldsymbol{x} \cdot \boldsymbol{y}}{|\boldsymbol{x}||\boldsymbol{y}|}. \tag{8}$$

**Sentence Delimitation** During generation time, a full candidate next sentence is considered generated if the language model has generated a new delimiter punctuation, i.e., a comma, period, question mark, or exclamation mark.

**C4-RealNews Preprocessing**    We separate the data points, which are paragraphs, into sentences using `nltk.sent_tokenize`. Additionally, we add a period mark to every sentence that does not end in a comma, period, question mark, or exclamation mark.

**Prompt for GPT-3.5-Turbo Paraphrase**    To use GPT-3.5-Turbo as a paraphraser, we provide the following prompt:

```
        Previous context:  {context} \n
   Current sentence to paraphrase:  {sent}
```

We define `sent` to be the target sentence to be paraphrased, and `context` as the list of sentences before the target sentence.

For the bigram paraphrase attack, we provide the following prompt:

```
Previous context:  {context} \n Paraphrase in {num-beams} different
          ways and return a numbered list :  {sent}
```

where `num-beams` specifies the number of candidate sentences. A higher `num-beams` will strengthen the bigram paraphrase attack but also at the cost of more computational resources.

