# OpenReview forum: "SemStamp: A Semantic Watermark with Paraphrastic Robustness for Text Generation"
_ICLR.cc/2024/Conference — ICLR 2024 Conference Withdrawn Submission_

### Official Review · Reviewer_feNQ · 2023-10-23

**Soundness:** 2 fair
**Presentation:** 3 good
**Contribution:** 3 good
**Rating:** 5
**Confidence:** 3

**Summary:**

In this paper, the authors introduce a novel semantic watermark for large language models named SemStamp. Only sentences that lie within a valid region of the feature space can be sampled during generation. Concurrently, they introduce a stronger paraphrasing scheme by generating sentences with minimal bigram overlap compared to the original. Experiments demonstrate that SemStamp exhibits greater robustness against both the baseline and the proposed paraphrase attacks than the baseline watermark.

**Strengths:**

- The paper is well-written and easy to comprehend.
- The concept is innovative, diverging significantly from the traditional token-level watermark. To my knowledge, this may be the pioneering work on semantic-level watermarking of large language models.
- The outcomes of the bigram-paraphrase attack are impressive, highlighting the efficacy of SemStamp.

**Weaknesses:**

- A primary concern of mine is the detection being at the sentence level. Consequently, I anticipate that SemStamp will exhibit a much lower z-score in comparison to Kirchenbauer et al. (2023a). This could pose issues when there are multiple keys/users, as there might be a need for correction during detection, such as the Bonferroni correction. Reliable detection requires the method to yield a notably low p-value. It would be beneficial if the authors could present some plots between the number of sentences and the z-scores/p-values.
- As the authors acknowledge, SemStamp demands significantly more generation time than preceding watermarks.

**Questions:**

- As I mentioned in the weaknesses section, could the authors provide some plots for the empirical z-score/p-value?
- The sentence encoder is trained on the C4-RealNews dataset, and the main experiments are conducted under the prompts from the same dataset. I am wondering how robust SemStamp is to some other unseen datasets.
- In practice, I think False Positive Rate = 5% sounds not pretty practical, could the authors show some numbers when the False Positive Rate is lower, like 1%?

---

> ### Author Response · Authors · 2023-11-18
> **Response to Reviewer feNQ**
>
> Dear reviewer feNQ,
>
> Thank you for your time conducting a holistic evaluation and providing us with precious insights! Here are our clarifications to the points you mentioned:
>
> ### **Z-score**
>
> Reviewer is very considerate in asking us to provide z-scores. The reason why we choose AUROC is that it measures the relative tradeoff between false positive rate and true positive rate. Various AI detection literature apply AUROC to evaluate the robustness of methods (Kirchenbauer et al 2023b, Mitchell et al 2023). Namely, while true positive texts have a lower z-score, false positive texts also exhibit a lower z-score, cancelling the low score effects. Therefore, AUROC already does the equivalent of something like Bonferroni Correction for us.
>
> ### **Robustness to unseen datasets**
>
> This is a great point! We made up experiments and please see our general response.
>
> ### **TP@FP=1%**
>
> Same as above, we’ve made up experiments and please see results in the general response section.
>
> We will be happy to address any additional concern you might have!

---

### Official Review · Reviewer_GzBM · 2023-10-28

**Soundness:** 1 poor
**Presentation:** 2 fair
**Contribution:** 1 poor
**Rating:** 3
**Confidence:** 5

**Summary:**

The paper proposes a watermarking for large language models. The idea behind the method is to watermark a semantic meaning instead of just words, as the prior art does. The watermark is embedded in transformed key-dependent domain, which is a classic trick in watermarking (see for example [1]). The embedding by rejection sampling is a novel feature, though the rejection sampler can have very poor efficiency and allows to embed very small watermark. While the text is not specific about the length of the watermark, I believe it is to  be equal to the number of LSH projections, which is equal to two in experimental settings (paragraph about speed on page 8). This means that the rejection sampler have high success to embed, but the watermarking scheme can suffer from a high false positive rate. This does not seem to be true, since the AUC seems to be high. I would welcome this to be better clarified.

The major problem of the watermarking scheme is that the effect of watermarking is "big". Since the watermark is hidden in the semantic domain, it effectively prevents generating answers with some semantic meaning. This gives the watermark robustness against paraphrasing, but in my opinion the degradation is so severe that the scheme will be useless in practice. This is visible in Figure 5, where the watermarked text is less useful then the non-watermarked.

Overall, I like the idea to hide the watermark in the transformed domain, this brings the work closer to methods used to watermark digital images. But the execution in the text domains is more difficult than shown in the presented text.

I would welcome if authors acknowledge of existence watermarking techniques predating large-language model boom, which exist and are apparently forgotten [2,3].

[1] Bas, Patrick, and François Cayre. "Natural watermarking: a secure spread spectrum technique for woa." International Workshop on Information Hiding. Berlin, Heidelberg: Springer Berlin Heidelberg, 2006.

[2] Topkara, Mercan, Cuneyt M. Taskiran, and Edward J. Delp III. "Natural language watermarking." Security, Steganography, and Watermarking of Multimedia Contents VII. Vol. 5681. SPIE, 2005.

[3] Atallah, Mikhail J., et al. "Natural language watermarking and tamperproofing." International workshop on information hiding. Berlin, Heidelberg: Springer Berlin Heidelberg, 2002.

**Strengths:**

As I have written above, I like the idea to hide the watermark in transformed key-dependent domain. This is classic in text watermarking.

**Weaknesses:**

* The effect of watermarking on semantic meaning of the text seems to me to be severe. This is inherent problem of the scheme and difficult to fix. I think this renders the scheme useless.
* The length of the watermaking seems to be small. Since the watermark is hidden by rejection sampler, the length of the watermark has to be sufficiently small to make the rejection sampler efficient, which at the same time makes the false positive rate high. Notice that practical watermarking scheme should have false positive rates in order 10^-5 or 10^-6, because most of text is watermarked. You should report detection accuracy at this rate. FPrate 5% means the user will be constantly flooded with false alarms. Again, this is inherent weakness of the scheme, which renders it practically useless.

**Questions:**

* What is the length of the watermark?
* How did you estimated false positive rate precisely?

---

> ### Author Response · Authors · 2023-11-18
> **Response to Reviewer GzBM**
>
> Dear reviewer GzBM,
>
> Thank you for time and feedback! Here are our clarifications:
>
> ### **Watermark Length**:
>
> Reviewer mentions several times about the length of watermark and the idea of embedding “small” watermark. However, SemStamp does not involve any “length of watermark”. It only embeds the 1-bit information that “this generation is watermarked”. SemStamp works by rejection sampling to sample a valid sentence. It treats sentence as the basic unit of watermarking, but there is no such concept as “length” in our scheme. We will dearly appreciate if the reviewer can further clarify what they mean by “length”.
>
> ### **Semantic Diversity**:
>
> Reviewer thinks because the watermark operates in semantic domain, it limits semantic diversity of generations. Regarding generation diversity, evaluation with rep-ngram and text entropy in Table 2 shows that SemStamp does not significantly curtail text diversity. The intuitive explanation here is that each valid semantic region is large enough to allow sufficient diversity.
>
> ### **Estimation of false positive rate**:
>
> To estimate, we also input 1000 human text generations during detection phase. We iterate through a range of z-scores and respectively find the percentage of human texts incorrectly classified as machine texts (thus false positives), fixing on the z score corresponding to the desired false positive rate.
>
> For evaluation of robustness under a smaller false positive rate, please see our general response where we show results under FPR=1%.
>
> ### **Acknowledging watermarking techniques predating LLM boom**
>
> We really appreciate reviewer for bringing great insights and being very considerate in recommending additional papers for us to acknowledge, which we will gratefully include in our manuscripts.

---

> > ### Comment · Reviewer_GzBM · 2023-11-21
> >
> > I think that your watermark is called in community 0-bit watermarking, since the watermark is either embedded or non-existing, hence it does not carry any information. The length of the watermark is the number of bits the watermark can carry. This is zero in your case, which makes a perfect sense.
> >
> > I do not agree with authors that semantic watermarking does not affect the diversity of generated text. If the watermark divides the semantic space into regions, that the semantic diversity has to be restricted. I cannot imagine this not to happen. I do think distribution of n-grams (or entropy) is a misleading measure of diversity, because it measures distribution of "groups of " characters, which is not correlated with a distribution of semantic meaning.
> >
> > I acknowledge reading the TP@FPR. How the 1000 sentences to estimate distribution of normal text were created? What was their semantic distribution?

---

### Official Review · Reviewer_ouFD · 2023-10-29

**Soundness:** 3 good
**Presentation:** 2 fair
**Contribution:** 3 good
**Rating:** 5
**Confidence:** 4

**Summary:**

In this work, the authors propose a semantic watermarking method for LLM. The paper first shows that the prior work (ICML 2023 best paper) is less robust against a novel paraphrase attack so-called bigram. Then, the authors present a method that can achieve a higher robustness against this attack.

**Strengths:**

1. This work addresses a timely topic of LLM watermarking. The proposed method aims to achieve higher robustness over the existing methods.

2. The authors provide empirical evaluation to support the claim that the proposed watermark is more robust.

**Weaknesses:**

1. The threat model is not clearly defined. Since the goal of watermarking is to protect the IP of model owner, if the owner suspects that there might be an IP breach, the owner would just verify the watermark. It is unclear how the paraphrase would be used in practice.

2. The main concern is the baseline method still achieves reasonably good performance against bigram. Thus, the need for the proposed method is not well motivated.

3. The improvement on the robustness also seems to be marginal.

4. The evaluation is limited. How about the robustness against fine-tuning, prompt-tuning, and watermark overwriting? The paper can also benefit from making the evaluation metrics of stealthiness, efficiency and robustness more clear.

**Questions:**

See Weaknesses.

---

> ### Author Response · Authors · 2023-11-18
> **Response to Reviewer ouFD**
>
> Dear reviewer ouFD,
>
> Thank you for the great insights and advice to our paper! Here are our clarifications:
>
> ### **Goal of watermarking**
>
> Our watermarking scheme is not aimed at intellectual property protection, but to facilitate the detection of machine-generated texts through embedding an imperceptible pattern during generation and attempting to reproduce this pattern during detection time. On the other hand, a malicious actor can conduct attacks through removing such watermark embedded during generations, preventing detectors to reproduce this pattern. One common attack is through paraphrasing generations on sentence-level to remove watermarks, which is what our algorithms are robust against.
>
> ### **Baseline against bigram-attacks**
>
> Reviewer thinks that “baseline method still achieves reasonably good performance” against bigram attacks. For instance, under parrot bigram paraphrase attack, the auroc of baseline drops from 0.915 (without bigram attack) to 0.875, and its true positive rate drops from 62.9% to 45.4% when the false positive rate is at 1% (TP@FP=1%), and from 78.7% to 65.8% on TP@FP=5%, which is arguably not a good performance.  Likewise, SemStamp’s improvement on robustness is quite significant. Under bigram-attacks by different paraphrasers, SemStamp on average has higher TP@FP=5% than the baseline by around 0.15, which improves by at least 20%.
>
> ### **Evaluation**
>
> We kindly ask the reviewer to clarify how “fine-tuning, prompt-tuning, and watermark overwriting” is related to our work. Since the focus of this paper is on developing a specific paraphrase-robust watermark, we do not assert general robustness against all types of attacks.
>
> In terms of efficiency, in Table 3, we measure the average number of sampled sentences to produce one valid sentence. For robustness, we studied SemStamp’s performance under different paraphrasers, different generation lengths, and different text domains (see general response section). For stealthiness, we’ve shown in Table 2 that SemStamp generations do not exhibit obvious qualitative differences from vanilla generations.

---

> > ### Comment · Reviewer_ouFD · 2023-11-23
> > **Thanks for the authors' response!**
> >
> > Thanks for the clarification. After considering other reviewers' comments, I plan to keep my rating.

---

### Author Response · Authors · 2023-11-18
**General Response and Additional Experiments**

Dear reviewers,

Thank you all for your time and efforts in reading our paper and providing helpful feedback! We post some additional experimental data in this section to address your common concerns. First, we show the true positive rate of detection when false positive rate is kept at 1%, since 5% false positive rate was thought to be too large. As shown below, SemStamp is still more resistant against both regular and bigram paraphrase attacks under this rate.

### **Experimental results with TP@FP=1%**

| paraphraser | baseline | SemStamp |
| --- | --- | --- |
| pegasus | 0.902 | 0.905 |
| pegasus-bigram | 0.708 | 0.861 |
| parrot | 0.629 | 0.689 |
| parrot-bigram | 0.454 | 0.714 |

### **Experiments on robustness to different datasets**
|                        |   AUROC |   FPR@1% |   FPR5% |   BERTScore |
|:-----------------------|--------:|---------:|--------:|------------:|
| Baseline-pegasus         |   0.976 |    0.939 |     0.955 |       0.716 |
| SStamp-pegasus       |  **0.984** |    **0.942** |     **0.987**  |       0.706 |
| Baseline-pegasus-bigram  |   0.958 |    0.76  |     0.887 |       0.699 |
| SStamp-pegasus-bigram |   **0.984** |    **0.926** |    **0.988** |       0.685 |
| Baseline-parrot          |   0.937 |    0.67  |     0.826 |       0.574 |
| SStamp-parrot        |   **0.956** |    **0.826** |     **0.965** |       0.564 |
| Baseline-parrot-bigram   |   0.92  |    0.564 |     0.741 |       0.578 |
| SStamp-parrot-bigram |   **0.957** |    **0.816** |     **0.965** |       0.572 |

We use the same setup in Table 1 for BookSum dataset (Kryściński et al. 2021), which is a collection of narrative text summarization. Results show that SemStamp is still more robust compared to the baseline.